# Role of the Maritime Continent in the remote influence of Atlantic Niño on the Pacific

Siying Liu [1,2], Ping Chang [3] ✉, Xiuquan Wan [1,2] ✉, Stephen G. Yeager [4] &
Ingo Richter [5]

Atlantic Niño, the dominant climate mode in the equatorial Atlantic, is known to remotely force a La Niña-like response in the Pacific, potentially affecting seasonal climate predictions. Here, we use both observations and large-ensemble simulations to explore the physical mechanisms linking the Atlantic to the Pacific. Results indicate that an eastward propagating atmospheric Kelvin wave from the Atlantic, through the Indian Ocean, to the Pacific is the primary pathway. Interaction of this Kelvin wave with the orography of the Maritime Continent induces orographic moisture convergence, contributing to the generation of a local Walker Cell over the Maritime Continent-Western Pacific area. Moreover, land friction over the Maritime Continent dissipates Kelvin wave energy, affecting the strength of the Bjerknes feedback and thus the development of the La Niña-like response. Therefore, improving the representation of land–atmosphere–ocean interactions over the Maritime Continent may be fundamental to realistically simulate Atlantic Niño's impact on El Niño-Southern Oscillation.

As seasonal climate forecasts continue to improve, the use of these forecasts has increased across a range of application sectors, including agriculture, health, water management and others[1]. The current skill and reliability of seasonal climate forecasts, however, leave much room for improvement[2]. Since the key paradigm for seasonal climate forecasts is El Niño-Southern Oscillation (ENSO)—the most prominent coupled atmosphere–ocean phenomenon on seasonal-to-interannual timescales[3], it is paramount that we continue to improve our understanding of ENSO dynamics and its predictability.

Although ENSO occurs in the tropical Pacific, modes of climate variability outside of the tropical Pacific contribute to and interact with ENSO, influencing its onset, evolution, and termination[4]. In particular, studies have shown that Atlantic Niño or Atlantic Zonal Mode[5]—a phenomenon in the equatorial Atlantic similar to ENSO, albeit weaker amplitude and more sporadic, can remotely force a La Niña-like response in the Pacific[6–16]. The unfolding 2021/22 La Niña event is

consistent with this Atlantic-forcing-Pacific premise: following the exceptionally strong Atlantic Niño in the boreal summer of 2021, a La Niña peaked in the Pacific during the winter of 2021 (Supplementary Fig. 1). However, the significance of the Atlantic remote influence on ENSO has been recently questioned[17,18], suggesting that there are still many unresolved issues, and thus a need for deepening our understanding of the mechanisms underlying the Atlantic-to-Pacific teleconnection.

Mechanisms of Atlantic Niño remotely forcing La Niña in the tropical Pacific have been described in terms of an anomalous Walker Circulation with an ascending branch over the equatorial Atlantic (EA) where SST is anomalously warm and a descending branch over the Western and Central Pacific (WP/CP) (See Fig. 9 of ref. [11]). However, this description is based on a fully developed response of the atmosphere to Atlantic Niño sea-surface temperature (SST) forcing and does not explain how the anomalous Walker Circulation is first developed, which

[1]Frontier Science Center for Deep Ocean Multispheres and Earth System (FDOMES) and Physical Oceanography Laboratory, Ocean University of China, Qingdao, China. [2]College of Oceanic and Atmospheric Sciences, Ocean University of China, Qingdao, China. [3]Department of Oceanography, Texas A&M University, Texas, USA. [4]National Center for Atmospheric Research, Boulder, Colorado, USA. [5]Application Laboratory (APL), Research Institute for Value-Added-Information Generation (VAiG), Japan Agency for Marine-Earth Science and Technology (JAMSTEC), Yokohama, Japan. ✉e-mail: ping@tamu.edu; xqwan@ouc.edu.cn

leaves a gap in our understanding of the fundamental mechanism in the Atlantic-to-Pacific teleconnection. Some of the specific questions remaining to be examined include: how does the anomalous Walker Circulation develop outside the EA? Where does the development start and how? What dynamical processes are responsible for the development? Understanding these questions can have important potential implications for improving ENSO simulations and predictions. This study focuses on exploring these questions by combining observational analyses with large-ensemble climate simulations.

## Results

### Remote response to Atlantic Niño

To identify the development of the atmospheric response to Atlantic Niño SST forcing in the observations, we performed a lag-regression analysis against the boreal summer ATL3 SST time series after linearly regressing out the preceding boreal winter Niño3.4 SST index to remove the remote forcing effect of ENSO on the tropical Atlantic ("Methods"). Figure 1a–f show regressed SST anomalies, wind anomalies at 1000 hPa, zonal and vertical wind anomalies, as well as zonal mass stream-function ("Methods") in boreal spring (lag = −1), summer (lag = 0), and fall (lag = +1). During spring, when Atlantic Niño begins to develop[19], there is a well-defined local response manifested as anomalous westerlies over the western EA. Outside the EA, however, the remotely forced response is first developed over the WP and MC, where significant anomalous easterlies are observed during spring. This result is not sensitive to the method of removing ENSO's influence on the tropical Atlantic (Supplementary Fig. 2). To the west (east) of the easterlies, there is enhanced (reduced) precipitation (Supplementary Fig. 3a), indicating that local anomalous Walker Cell begins to form over WP and MC in response to the developing Atlantic Niño in EA during boreal spring (Fig. 1d). A further analysis indicates that the precipitation anomalies are primarily caused by circulation anomaly induced moisture convergence/divergence, rather than by local evaporation changes (Supplementary Fig. 3a, d, g, j). As the season progresses from spring to summer, Atlantic Niño reaches its peak, and the Atlantic Walker Cell is fully developed, while both the easterlies and

the equatorial cooling in the Pacific intensify in accordance with the Bjerknes feedback[20]. The upwelling and the surface cooling excited by the easterlies expand eastward, leading to the further development of the anomalous Walker Cell over the WP. In the following fall and winter, as Atlantic Niño is winding down, the positive Bjerknes feedback continues in the tropical Pacific, eventually giving rise to a full-fledged La Niña (Fig. 1c and Supplementary Fig. 4a) coupled with a fully developed anomalous Walker Circulation over the Pacific (Fig. 1f). This depiction of the development of the Atlantic Niño teleconnection is consistent with previous studies[6,9,16].

The first appearance of the easterlies over the WP and MC, along with the westerlies over the western EA as the initial response to Atlantic Niño SST, is reminiscent of the classic Matsuno–Gill response to diabatic heat source in the EA[21], with a fast, eastward propagating atmospheric Kelvin wave, carrying easterly wind anomalies across the Indian Ocean into the Pacific. This Kelvin wave mechanism linking Atlantic Niño to the Pacific has been proposed[14,22], but not fully validated in a coupled modeling framework. To further validate this mechanism, we conducted large-ensemble Community Earth System Model[23] (CESM) simulations (hereafter CTRL), where an observed Atlantic Niño SST anomaly is prescribed (Supplementary Fig. 5 and "Methods"). Figure 2a–c shows the simulated SST and winds at 992 hPa anomalies during spring, summer, and fall. It is evident that the simulation captures the salient feature of the observations during the development of Atlantic Niño teleconnections. In particular, the concurrent westerly anomalies over the western EA and easterly anomalies over the WP and MC are well simulated, except that the response is stronger than observed. This is because the large-ensemble experimental design reduces the influence of atmospheric internal variability through ensemble averaging, allowing for better detection of the Kelvin wave signal. The persistent Atlantic Niño SST pattern used in the simulations also contributes to the enhanced remote response. It is important that CTRL can realistically reproduce the observed Atlantic-to-Pacific teleconnection because it provides not only support to the observational analysis but also a reference for the mechanistic model simulations described below.

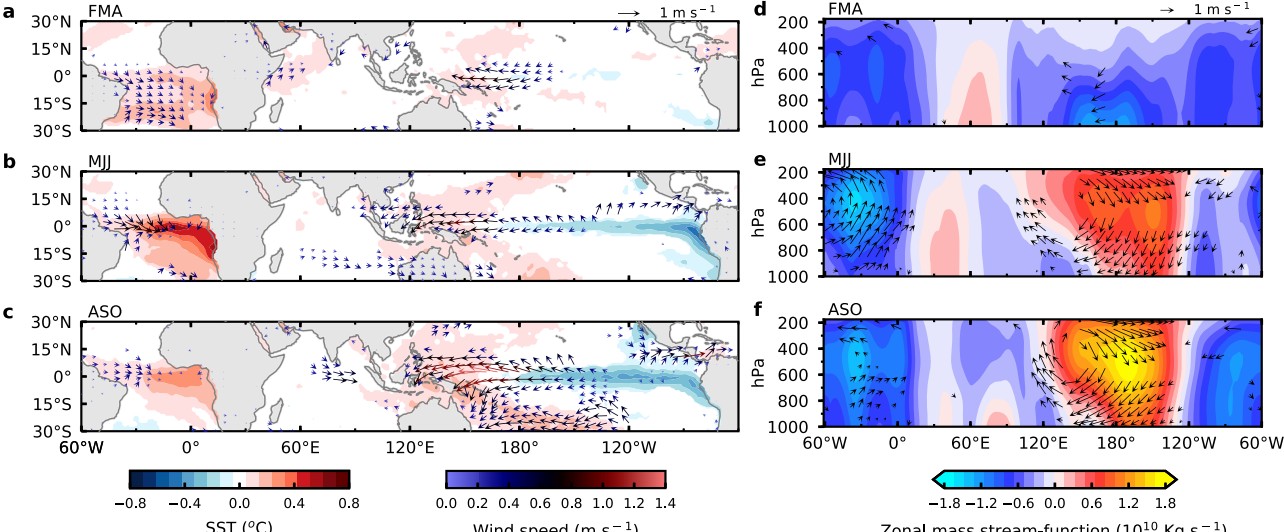

**Fig. 1 | Observed tropical Sea Surface Temperature (SST) and Walker Circulation response to Atlantic Niño. a–c** Boreal spring (February-March-April (FMA)) (**a**), summer (May-June-July (MJJ)) (**b**), and fall (August-September-October (ASO)) (**c**) SST (°C; color shading) anomalies (HadISST datasets, 1959–2021) and wind anomalies at 1000 hPa (m s⁻¹; color arrows) averaged using three reanalysis datasets (ERA5, JRA55 for 1959–2021 and ERA-Interim for 1979–2018) and regressed onto the normalized May-June-July (MJJ) Atlantic Niño-related variability (ATL3 index) after linearly regressing out El Niño-Southern Oscillation related variability ("Methods").

**d–f** Same as (**a–c**) except for zonal and vertical wind anomalies (m s⁻¹; arrows; vertical winds multiplied by 100) averaged between 5°S and 5°N, and zonal mass stream-function anomalies (Kg s⁻¹; color shading; see "Methods") from 175 to 1000 hPa pressure levels. The colors (**a–c**) and vectors (**a–f**) indicate statistical significance at the 10% level based on a two-sided Student's t-test. The statistical test was not performed for the zonal mass stream-function because its main purpose is to illustrate the direction of the zonal circulation anomalies. Source data are provided as a Source Data file in the Zenodo database.

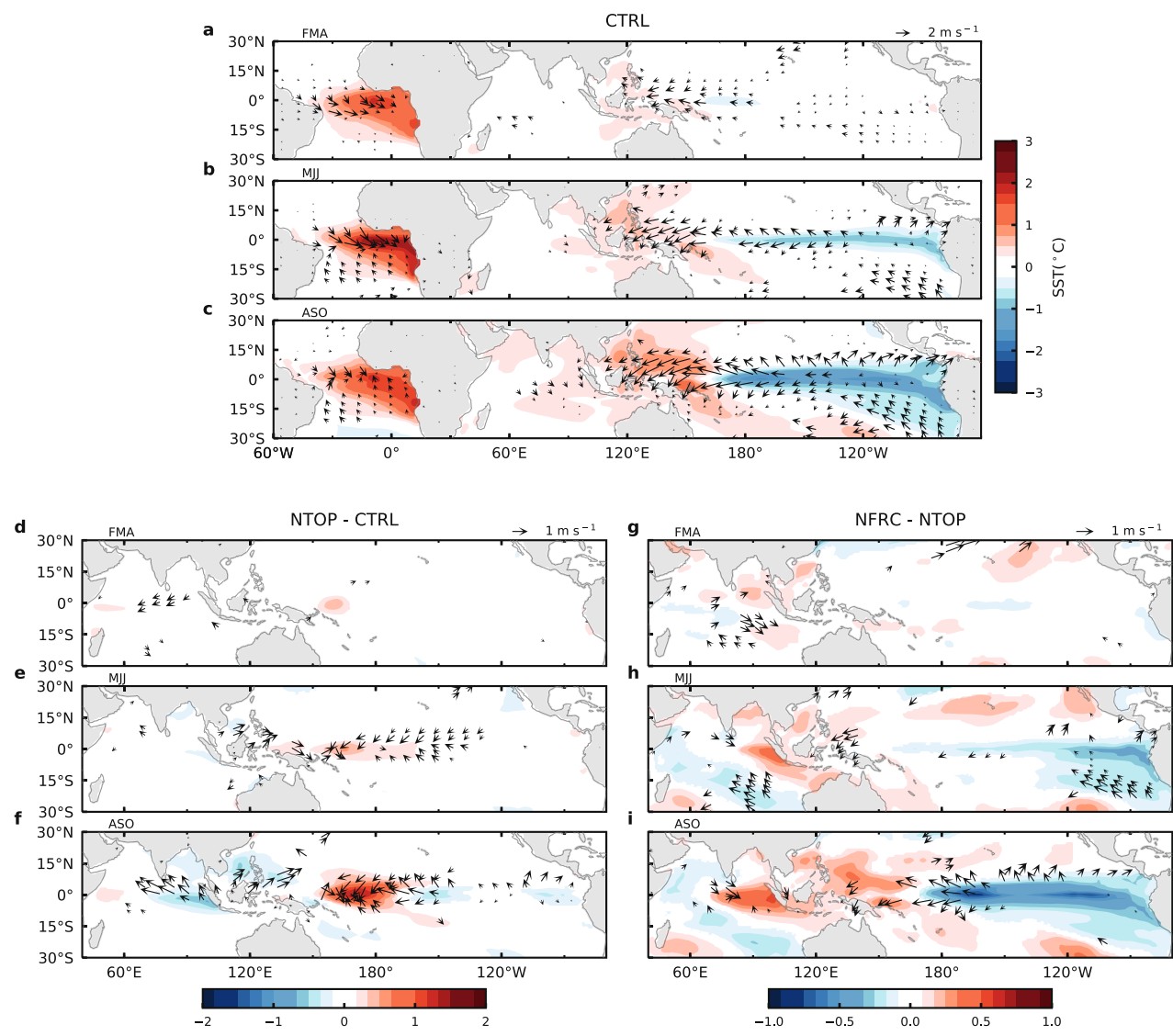

**Fig. 2 | Simulated response of Sea Surface Temperature (SST) and low-level wind anomalies to Atlantic Niño SST in three large-ensemble experiments.** **a**–**c** Simulated SST (°C; color shading) and wind anomalies at 992 hPa (m s⁻¹; arrows) averaged over boreal spring (February-March-April (FMA)) (**a**), summer (May-June-July (MJJ)) (**b**) and fall (August-September-October (ASO)) (**c**) in the control experiment (CTRL) described in the text; **d**–**f** Same as (**a**–**c**) except for the anomalies in the topography removal experiment (NTOP) described in the text minus those in CTRL; **g**–**i** Same as (**a**–**c**) except for the anomalies in the land-friction reduction experiment (NFRC) described in the text minus those in NTOP.

Anomalies in CTRL, NTOP and NFRC are obtained by taking the ensemble-mean of CTRL with positive Atlantic Niño SST anomalies forcing minus CTRL with negative Atlantic Niño SST anomalies forcing (CTRL+ minus CTRL−), ensemble-mean of NTOP+ minus NTOP−, and ensemble-mean of NFRC+ minus NFRC−, respectively. Colors and vectors represent those that are significant at the 10% level based on a two-sided Student's t-test. Note that in (**d**–**f**) and (**g**–**i**), the tropical Atlantic sector is excluded because the forcing is the same for all three experiments. Source data are provided as a Source Data file in the Zenodo database.

It is worth noting that although the simulated SST in the tropical Atlantic is heavily constrained by the prescribed time-invariant Atlantic Niño SST anomalies, there are still weak variations in equatorial Atlantic SST with a peak during boreal summer in agreement with the observations[19]. This indicates that the unconstrained subsurface ocean variability in response to Atlantic Niño SST-induced trade wind changes can modulate the simulated equatorial Atlantic SST (Supplementary Fig. 7) through local Bjerknes feedback within the tropical Atlantic. This local Bjerknes feedback in the Atlantic may contribute to the enhanced remote response in the Pacific during boreal summer.

The simulated ensemble-mean geopotential height anomalies at 200 hPa clearly show a Matsuno–Gill type response to the heating over the EA: a faster eastward propagating Kelvin wave to the east of the heating and a slower westward propagating Rossby wave to the

west of the heating (Supplementary Fig. 8). Consistent with the development of the easterly anomalies, the geopotential height anomalies arrive at the WP and MC from the west within the first two months after the switch-on of the Atlantic SST forcing (Supplementary Fig. 8a) and become quasi-stationary over the WP and MC while building up strength in the following three months (Fig. 3a–c). The zonal circulation averaged between 5°S and 5°N along the equator shows a well-defined anomalous Walker Cell between 100°E and 180°E with an ascending branch in the west and a descending branch in the east (Fig. 4g). The simulated precipitation anomalies are also consistent with the anomalous Walker Cell (Supplementary Fig. 9a–c). In the following months of the anomalous Walker Cell development, there is a build-up of negative ocean heat content anomalies in the eastern equatorial Pacific (Supplementary Fig. 6a–c) along with an intensifying cold SST anomaly and a strengthening

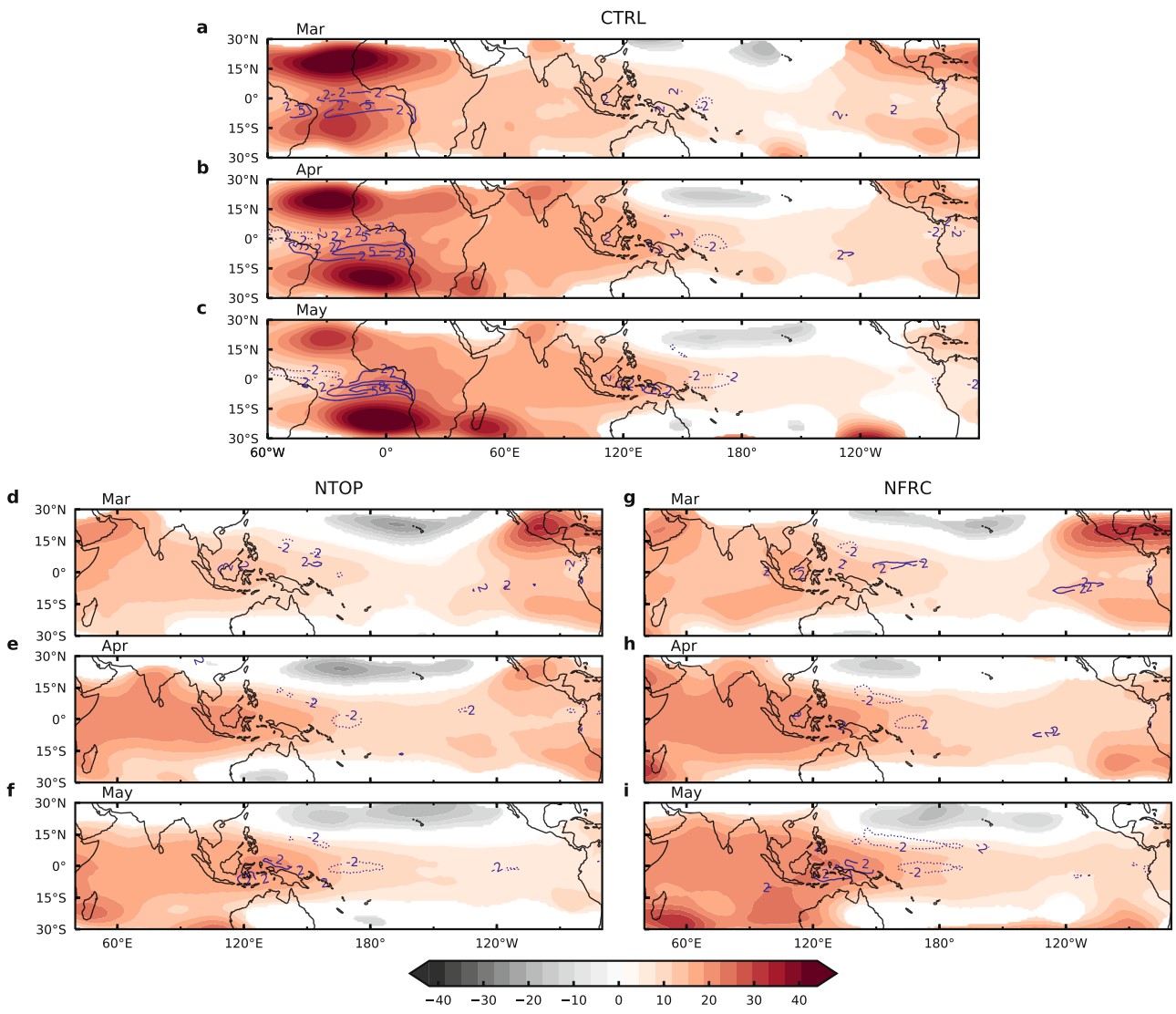

**Fig. 3 | Simulated atmospheric Kelvin wave-like response and precipitation anomalies in three large-ensemble experiments. a–c** Simulated geopotential height anomalies at 200 hPa (m; color shading) and precipitation anomalies (mm day$^{-1}$; contour lines; contour interval 3 mm day$^{-1}$) in March (Mar) (**a**), April (Apr) (**b**), May (**c**) for the control experiment (CTRL) described in the text. **d–f** Same as (**a–c**) except for the topography removal experiment (NTOP) described in the text. **g–i** Same as (**a–c**) except for the land-friction reduction experiment (NFRC) described in the text. Anomalies in CTRL, NTOP and NFRC are obtained by taking the ensemble-mean of CTRL with positive Atlantic Niño SST anomalies forcing minus CTRL with negative Atlantic Niño SST anomalies forcing (CTRL+ minus CTRL−), ensemble-mean of NTOP+ minus NTOP−, and ensemble-mean of NFRC+ minus NFRC−, respectively. Mar-May anomalies are selected to show the well-developed Kelvin wave-like response over the Maritime Continent and western Pacific, and the initial eastward propagating wave signal is shown in Supplementary Fig. S5. Only the precipitation anomalies larger than 2 mm day$^{-1}$ (solid contour lines) and smaller than −2 mm day$^{-1}$ (dashed contour lines) are shown by contour lines. Colors and contour lines represent those that are significant at the 10% level based on a two-sided Student's *t*-test. As in Fig. 2, the tropical Atlantic sector is excluded in (**d–f**) and (**g–i**). Source data are provided as a Source Data file in the Zenodo database.

easterly anomaly over the CP and Eastern Pacific (EP). This is indicative of a basin-wide Bjerknes feedback, which leads to the full development of La Niña during boreal winter (Supplementary Fig. 10a). Therefore, the large-ensemble CESM simulation not only supports the Kelvin wave mechanism but also reveals the key role of the local Walker Cell over the WP and MC in initiating the development of the Atlantic-to-Pacific teleconnection.

The Atlantic Niño SST forcing also generates a Rossby wave component of the Matsuno–Gill solution that carries a weak westerly wind anomaly westward (in contrast to the easterly anomaly of the Kelvin wave) (Supplementary Fig. 8). During boreal spring, brief warming in the far eastern equatorial Pacific may be associated with this Rossby wave (Fig. 2a). But apparently this forcing is too weak to compete with the easterly anomalies induced by the Kelvin wave and

the subsequent Bjerknes feedback. The initial warming appears to be quickly overcome by the strong equatorial cooling associated with the easterly wind anomaly developed in the wake of the Kelvin wave. One possible reason for the weak Rossby wave forcing is that the wind anomalies associated with the Rossby wave have maximum amplitude off the equator (Figs. 1b and 2b) and thus are not effective in triggering Bjerknes feedback.

The above analyses raise the question of why and how the local Walker Cell is initiated over the WP and MC. We hypothesize that the generation of the local Walker Cell may be related to the interaction between the Kelvin wave and MC landmass. First, orography over the MC may interact with anomalous moisture transport induced by the Kelvin wave, producing orography-induced moisture convergence/divergence[24,25], which can then trigger local convection over the MC.

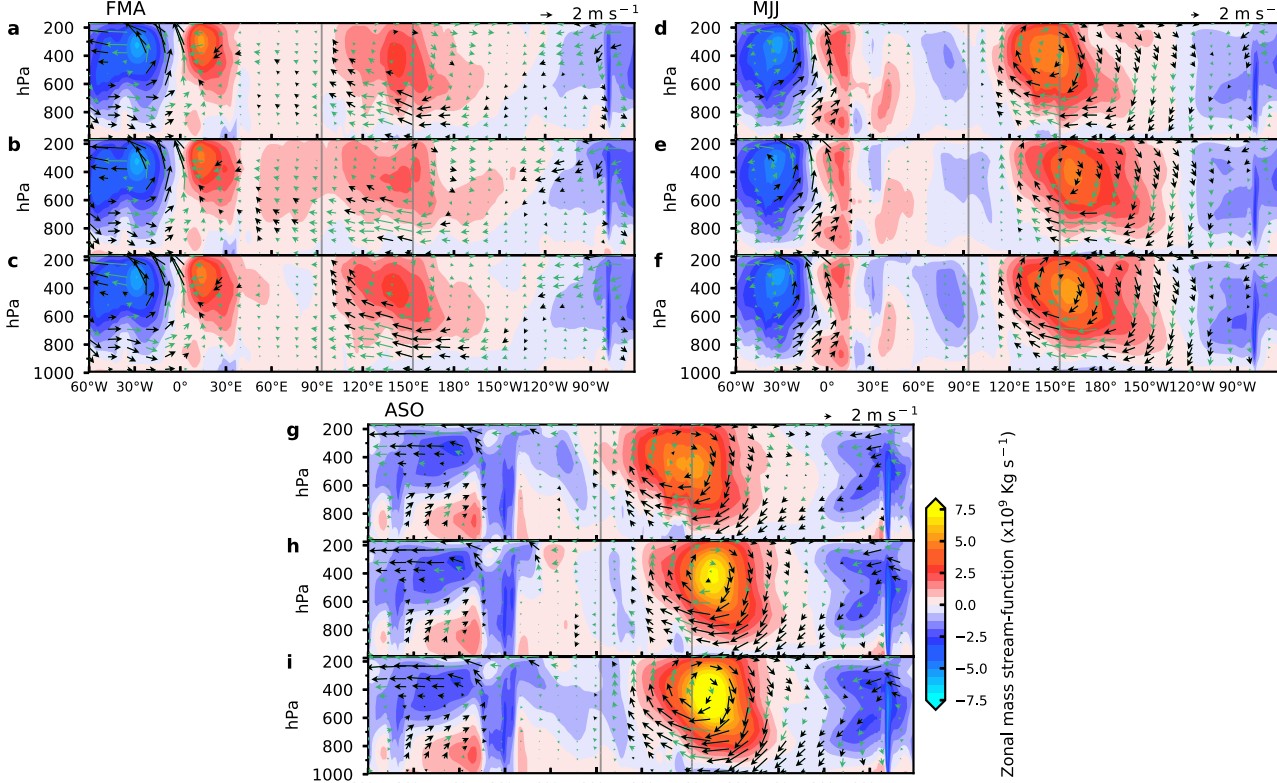

**Fig. 4 | Simulated anomalous Walker Circulation in response to Atlantic Niño Sea Surface Temperature (SST) in the three large-ensemble experiments.** **a–c** Simulated zonal circulation anomalies (m s⁻¹; arrows; vertical winds multiplied by 100) and zonal mass stream-function anomalies (Kg s⁻¹; color shading) averaged between 5°S and 5°N along the equator from 175 to 1000 hPa pressure levels for boreal spring (February-March-April (FMA)) in the control experiment (CTRL) described in the text (**a**), the topography removal experiment (NTOP) described in the text (**b**), and the land-friction reduction experiment (NFRC) described in the text (**c**); **d–f** Same as (**a–c**) except for summer (May-June-July (MJJ)); **g–i** Same as (**a–c**) except for fall (August-September-October (ASO)). Black (green) arrows indicate those that are (not) significant at 10% level based on a two-sided Student's t-test. Anomalies in CTRL, NTOP and NFRC are obtained by taking the ensemble-mean of CTRL with positive Atlantic Niño SST anomalies forcing minus CTRL with negative Atlantic Niño SST anomalies forcing (CTRL+ minus CTRL−), ensemble-mean of NTOP+ minus NTOP−, and ensemble-mean of NFRC+ minus NFRC−, respectively. The Maritime Continent sector is illustrated between the two gray lines. Source data are provided as a Source Data file in the Zenodo database.

Second, the land friction over the MC may increase the dissipation of the Kelvin wave, arresting the eastward propagation of the Kelvin wave near the MC. The accumulated easterly anomalies of the Kelvin wave near the MC can then cause cooling in the WP through enhancing equatorial upwelling, which in turn further intensifies and expands the easterlies via Bjerknes feedback, creating subsidence over the region. As such, these land processes over the MC may play an important role in the development of the local Walker Cell. To test this hypothesis, we conducted two additional large-ensemble experiments.

**Role of the Maritime Continent**
In the first ensemble (hereafter NTOP), we removed all the topographic features over the MC (93°E–152°E, 10°S–10°N) while keeping everything else the same as CTRL ("Methods" and Table 1). This experiment allows us to examine the interaction between the Kelvin wave and MC orography. In the second ensemble (hereafter NFRC), in addition to removing all the MC topographic features, we further reduced the land friction over the MC by modifying the momentum roughness length ("Methods"). This experiment allows us to study the importance of the MC land friction effect on the Kelvin wave forced by Atlantic Niño SST.

To examine the role of MC orography, Fig. 2d–f contrast the responses of SST and winds at 992 hPa between CTRL and NTOP. During the boreal spring, flattening orography results in negative precipitation anomalies over the MC (Supplementary Fig. 9d) and significantly warmer SST anomalies east of it, suggesting a slower build-up and eastward-shifted local anomalous Walker Cell over the

MC and WP (Fig. 4). This eastward-shifted Walker Cell becomes more evident in the boreal fall in the simulation when it is more developed (Fig. 4h). Compared to CTRL (Fig. 4g), the Walker Cell in NTOP shifts eastward by about 20° (Fig. 4h). Along with the eastward-shifted Walker Cell, the cold SST anomaly in the equatorial Pacific also shifts eastward (Fig. 2d–f). The further eastward expansion of the Kelvin wave due to the removal of MC orography can also be seen in geopotential height anomalies (Fig. 3d–f and Supplementary Fig. 11a–c) in the following three months (March-April-May) after the signal becomes quasi-stationary.

While flattening MC orography in NTOP shows a clear impact on the local Walker Cell over the MC and WP, reducing land friction over the MC in NFRC shows a more significant impact on the strength of the La Niña-like response in the Pacific to the Atlantic forcing. As expected, weakening the land friction over the MC causes the Kelvin wave to propagate further into the central and eastern Pacific in NFRC (Fig. 3g–i and Supplementary Fig. 11d–f), producing stronger easterly anomalies (Fig. 2g–i), which triggers stronger Bjerknes feedback along the equator in the CP and EP (Supplementary Fig. 6h, i) and gives rise to a strengthened La Niña-like response (Fig. 2h, i). The enhanced and further eastward extended low-level winds and their interactions with the underlying SST in NFRC are consistent with the stronger Walker Cell (Fig. 4f, i). We note that the analysis used here to compare these different ensembles implicitly assumes weak nonlinearity in the system, such that the climatological state changes due to the modification of the MC orography and land friction in NTOP and NFRC do not

**Table 1 | Simulations with Atlantic Niño sea surface temperature anomalies forcing**

| Simulation name | Tropical Atlantic | Maritime Continent (Land grid-point) | Ensemble members | Integration length |
|---|---|---|---|---|
| CTRL+ | Model climatology sea surface temperature (SST) plus Atlantic Niño SST anomalies | Realistic topography | 60 | 1 year |
| NTOP+ | | Flattened topography | | |
| NFRC+ | | Flattened topography and reduced land friction | | |
| CTRL− | Model climatology SST minus Atlantic Niño SST anomalies | Realistic topography | | |
| NTOP− | | Flattened topography | | |
| NFRC− | | Flattened topography and reduced land friction | | |

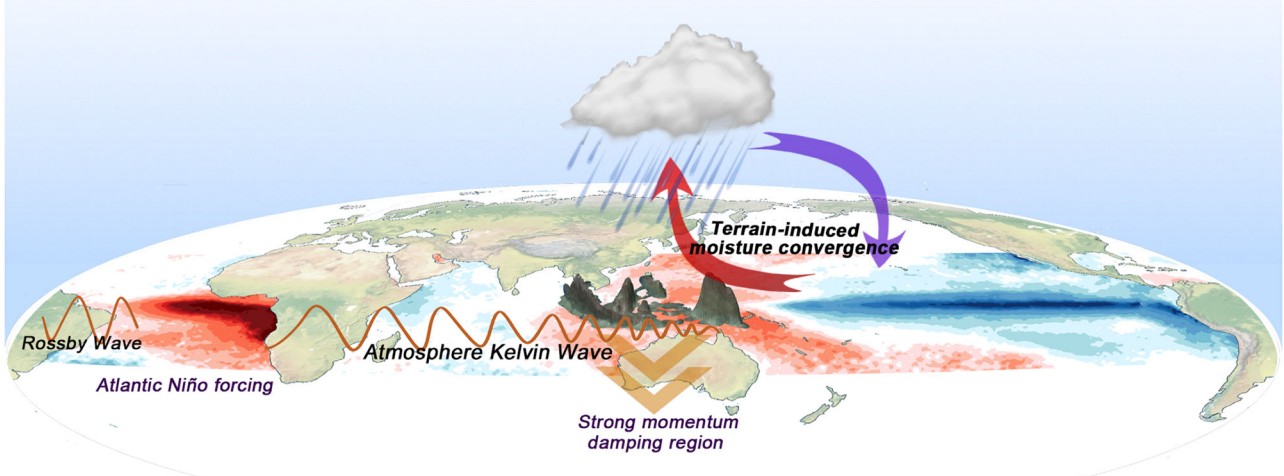

**Fig. 5 | Schematic diagram for Atlantic-to-Pacific teleconnection.** In response to an Atlantic Niño Sea Surface Temperature (SST) forcing, an eastward propagating atmospheric Kelvin wave carries easterly anomalies to the Maritime Continent (MC) and Western Pacific (WP) through the Indian Ocean. The easterly anomalies interact with the MC orography and generate terrain-induced moisture convergence that produces a local Walker Cell. Meanwhile, the MC land friction dissipates Kelvin wave energy and helps to anchor the local Walker Cell over the MC and WP. The local Walker Cell subsequently interacts with the ocean in the WP through Bjerknes feedback, which leads to the development of a La Niña-like response in the Pacific.

introduce strong nonlinear effects on the response to Atlantic Niño SST. We present additional analyses in the Supplementary Discussion that validate the weak nonlinearity assumption. Therefore, these numerical experiments support our hypothesis that the interaction between the Kelvin wave and MC landmass plays a critical role in the Atlantic-to-Pacific teleconnection.

The above results are summarized in a schematic diagram in Fig. 5: a warm SST anomaly in the equatorial Atlantic invigorates the tropical convection and excites an eastward propagating Kelvin wave whose easterly anomalies interact with the MC landmass, initiating the development of a local Walker Cell over the WP, which then triggers Bjerknes feedback along the equator in the WP, leading to the development of a La Niña event. The atmosphere–land–ocean interaction over MC can regulate the position and strength of the La Niña-like response. In the absence of this interaction, the La Niña-like response is likely to be stronger. Therefore, the representation of atmosphere–land–ocean interaction over the MC in climate models may have a direct impact on the Pacific response to Atlantic Niño.

## Discussion

In this study, we advance the fundamental understanding of mechanisms linking equatorial Atlantic SST variability to Pacific ENSO by first confirming the Kelvin wave pathway as the key mechanism linking Atlantic Niño to Pacific ENSO and then demonstrating the importance of atmosphere–land–ocean interactions over the MC in regulating the position and strength of the Pacific response to the Atlantic forcing. The finding that the terrain and land surface processes over the MC can

affect the development and strength of the Pacific response indicates that these processes can potentially have an impact on ENSO prediction. As such, there is a need to further understand and improve the representation of atmosphere–land–ocean interactions over the MC in climate models. Most current-generation climate models, including the one used in this study, have coarse horizontal resolutions of ~100 km. At this resolution, orography is not well represented by the models. Therefore, it is possible that the interaction between the Kelvin wave and MC orography may be underestimated, which may further influence model simulations of ENSO. Future studies are needed to examine how increasing model resolution can affect the atmosphere–land–ocean interaction over the MC, as it is a critical part of inter-basin interaction between the equatorial Atlantic and Pacific.

We emphasize that the focus of this study is on the underlying mechanism of the Atlantic-to-Pacific teleconnection, but not on the debate of how significant an impact Atlantic Niño has on ENSO and its predictability[17,18]. However, the results presented here do indicate that a poorly resolved MC orography may have a direct impact on simulations of the Pacific response to the Atlantic forcing that may be important to improving ENSO-related climate prediction. To fully address Atlantic Niño impact on ENSO predictability, a multi-model approach with a common experimental design is highly desirable because such an approach can allow for a comprehensive evaluation of model bias effects and sensitivity to model physics and resolution, as well as a robust quantification of the impact of Atlantic Niño on ENSO prediction. The results of this study provide further motivation for a future multi-model intercomparison study on Atlantic-to-Pacific teleconnection.

## Methods

### Lag-regression analysis based on observations

In Fig. 1 and Supplementary Fig. 3, partial lag-regressions[6,10] were performed with respect to the normalized May-June-July (MJJ) ATL3 index, which was defined by averaging detrended equatorial Atlantic SST anomalies over the region 20°W–0° and 3°S–3°N, after linearly regressed out December-January-February (DJF) Niño3.4 index from all the variables to remove ENSO's influence on Atlantic Niño. Here the Niño3.4 index was defined by averaging equatorial Pacific detrended SST anomalies over the region 170°W–120°W and 5°S–5°N, and both the ATL3 and ENSO indices were normalized by dividing by their respective standard deviations. Note that we did not use the commonly used Jun-Jul-Aug (JJA) ATL3 index in Fig. 1 because Atlantic Niño-related SST anomalies reach the maximum in the ATL3 region during June based on a composite analysis of observed Atlantic Niño events[19]. The partial linear lag-regression against the Feb-Mar-Apr (FMA) ATL3 index is also shown in Supplementary Fig. 12 after regressing out the DJF Niño3.4 index. It shows positive SST regression coefficients over much of the equatorial and south tropical Atlantic, along with a westerly anomaly in the western tropical Atlantic during FMA. Over the western Pacific, a strong easterly anomaly coupled with some weak SST anomalies is also detected during FMA. These regression patterns exhibit strong similarity with the regression against the MJJ ATL3 index in Fig. 1, except that the regressed SST anomalies in the tropical Atlantic decay faster in amplitude, as expected.

To test the robustness of the results, we also used a different method to remove ENSO's influence, in which we first excluded the years of strong ENSO from the data record and then performed lag-regression analysis against the MJJ ATL3 index. A strong ENSO year was defined from current January to December if the $D^{-1}J^0F^0$ (−1 indicates the previous year and 0 indicates the current year) Niño3.4 index exceeded its standard deviation. The results are shown in Supplementary Fig. 2 and are similar to those in Fig. 1, indicating that the regression patterns are not sensitive to how ENSO is removed.

The datasets used include the monthly SST at 1°×1° horizontal resolution from the UK Met Office Centre HadISST datasets[26] during 1959–2021, and the winds at various isobaric levels from 175 to 1000 hPa from ERA5 Reanalysis[27] (1959–2021), ERA-Interim Reanalysis[28] (1979–2018) and JRA55 Reanalysis[29] (1959–2021). To fully utilize the three reanalysis products and minimize internal variability, we first averaged the wind fields from the three reanalysis datasets and then performed the lag-regression analysis (Fig. 1 and Supplementary Figs. 2 and 10). Since the time periods of the reanalysis products are different, we used all three datasets in the common time period of 1979–2018 and only used ERA5 and JRA-55 datasets for the remaining periods to derive the averaged wind fields. In addition, total precipitation, vertical integral of the divergence of moisture flux (referred to as Div(Uq)), vertically integrated moisture divergence (referred to as qDivU) and evaporation from ERA5 Reanalysis for 1959–2021 were also used to derive the results in Supplementary Fig. 3. After excluding the preceding strong ENSO years, there are 44 years (January–December) for lag-regression analyses from the HadISST datasets and averaged reanalysis datasets (Supplementary Table 2).

### Large-ensemble pacemaker experiments

All the large-ensemble experiments were conducted using Community Earth System Model[23] version1.1.2 (CESM1.1.2) with a nominal -1° horizontal resolution for all model components that include ocean, atmosphere, land, sea ice, land ice, and river components. The ocean component model−Parallel Ocean Program version 2 (POP2) has 60 vertical levels, the atmosphere component model−Community Atmospheric Model version 5.2 (CAM5.2) adapts 30 hybrid vertical pressure levels, and the land component model−Community Land Model version 4 (CLM4) shares the same horizontal grid with atmosphere component, while the sea-ice model−Community Ice Code

version 4 (CICE4) shares the same horizontal grid with the ocean component. Three ensembles of the CESM1.1.2 simulations were performed, each of which consists of two sets of simulations−one with positive (+) Atlantic Niño SST anomaly (Supplementary Fig. 5a) imposed on model daily-mean SST climatology and another with negative (−) Atlantic Niño SST anomaly (Supplementary Fig. 5b) imposed on the model daily-mean SST climatology. During the entire integration, both the positive and negative SST anomalies were kept constant. Each set consists of 60 ensemble members for 1-year simulations with slightly different atmospheric initial conditions (i.e., adding small perturbations to an atmospheric restart file taken from January 1 of the year 2006 of a long pre-industrial control run)[23]. All ensemble members were initialized with the same ocean condition with a neutral ENSO state in the tropical Pacific on January 1 of the year 2006, of the same pre-industrial control simulation. Pre-existing ENSO states in the tropical Pacific may affect the Atlantic remote influence mechanism, so we chose a neutral ENSO year as the ocean's initial condition for the ensembles. The three different ensembles differ in modifications to the MC land surface. The control ensemble (referred to as CTRL+/−) has no modifications made to MC and follows the standard pacemaker experiment design[15], in which model SST is restored to the prescribed +/− Atlantic Niño SST plus the daily-mean model climatology over the region 69°W–18°E and 24°S–24°N with a restoring timescale of 5 days and 8° of buffer zones in the meridional direction at the northern and southern boundaries (Supplementary Fig. 5). The MC orography removed ensemble (referred to as NTOP+/−) has the same forcing configuration in the tropical Atlantic, but with flattened orography over the region 93°E–152°E and 10°S–10°N. The MC land friction reduced ensemble (referred to as NFRC+/−), in addition to the change in NTOP+/−, reduces land friction by setting the momentum roughness length to $1 \times 10^{-10}$ m over the MC land surface (see more discussion in the next session in "Methods"). A brief summary of the three ensembles is given in Table 1.

Since we combined the observed Atlantic Niño SST anomaly with model climatology, the model SST bias is retained in the simulations. The advantage of such a setup is that model mean states in each tropical basin are dynamically consistent, but the disadvantage is that the model bias may have an impact on simulating Atlantic Niño teleconnection. However, CESM1 biases are relatively small with SST bias of less than ±1 °C over much of the tropical oceans (Supplementary Fig. 13). Also, because we take the difference between the positive (+) and negative (−) Atlantic Niño SST forcing sets to analyze Atlantic Niño teleconnection (e.g., CTRL+− CTRL−), the model biases tend to cancel out each other assuming the response behaves linearly. Even if the response is nonlinear, we expect the effect of the model bias to be reduced significantly by taking the difference between positive and negative SST forcing simulations. We believe this experimental design is well suited to study the mechanism of Atlantic-to-Pacific teleconnection, even though the forcing is idealized. The potential nonlinearity in response to Atlantic warm and cold SST anomalies forcing is assessed in the Supplementary Discussion.

### Atlantic Niño (Niña) forcing

The Atlantic Niño SST forcing pattern (Supplementary Fig. 5) used for the large-ensemble experiments was obtained by regressing the detrended observed SST anomalies from HadISST onto the JJA ATL3 index for the period of 1870–2017. The same pattern is used for CTRL-, NTOP-, and NFRC- with the SST anomaly multiplied by −1. The same Atlantic Niño SST forcing pattern was superimposed on model climatological daily-mean SST in the tropical Atlantic between 24°S and 24°N in the three sets of experiments.

### Land surface layer momentum fluxes in CLM4

Momentum fluxes over land grid points are determined using sub-grid roughness lengths (m) and displacement heights (m) appropriate for

the land surface heterogeneity. The momentum fluxes between the land surface and atmospheric reference height are written in the form[30]:

$$\boldsymbol{\tau} = -\rho_{atm}\frac{(\mathbf{u}_{atm} - \mathbf{u}_s)}{r_{atm}}, \qquad (1)$$

where $\mathbf{u}_{atm}$ is the horizontal wind vectors at the reference height, $\mathbf{u}_s$ is the horizontal wind vectors at height $Z_{0m} + d$, where $Z_{0m}$ is the roughness lengths for momentum (m), and $d$ is the displacement height (m). $\rho_{atm}$ is the density of atmospheric air (Kg m$^{-3}$), $r_{atm}$ is the aerodynamic resistance to momentum transfer (s m$^{-1}$).

Based on the Monin–Obukhov similarity theory[31], the dimensionless mean horizontal wind speed profile gradients depend on the similarity function of $\zeta = \frac{Z-d}{L}$[32], where $Z$ is the atmosphere height (m), and $L$ is the Monin–Obukhov length scale (m) that accounts for buoyancy effects resulting from vertical density gradients (i.e., the atmospheric stability). The momentum aerodynamic resistances can then be written as[30]:

$$r_{atm} = \frac{1}{k^2 V_a}\left[\ln\left(\frac{Z_{atm}-d}{Z_{0m}}\right) - \psi_m\left(\frac{Z_{atm}-d}{L}\right) + \psi_m\left(\frac{Z_{0m}}{L}\right)\right]^2, \qquad (2)$$

$$\psi_m(\zeta) = \int_{Z_{0m}/L}^{\zeta}\frac{[1-\phi_m(x)]}{x}dx, \qquad (3)$$

where $k$ is the von Kármán constant (0.4), $V_a$ is the wind velocity at reference height, including the convective velocity, and $\phi_m$ is a function of $\zeta$ that relates the constant fluxes of momentum to the mean profile gradients of winds.

In CLM4, $Z_{0m}$ is a constant (0.01 m) for soil, glaciers and wetland non-vegetated surfaces. For vegetated surfaces, $Z_{0m}$ depends on plant height and adjusts for canopy density ($10^{-1}$–$10^{1}$ m)[30]. In NFRC simulation of this study, $Z_{0m}$ is set to an artificially small value of $1 \times 10^{-10}$ m to reduce land surface friction.

## Zonal mass stream-function

The intensity of the Walker Circulation can be estimated by the zonal mass stream-function[33,34] (Kg s$^{-1}$), which can be calculated by:

$$\Psi = \frac{a\triangle\varphi}{g}\int_0^p u_D dp, \qquad (4)$$

where $a$ is the radius of the Earth, $\Delta\varphi$ is the width of 10°S–10°N in radians proportional to Earth's radius, $g$ is the gravity constant, and $u_D$ is the divergent part of zonal winds averaged between 5°S–5°N in the meridional direction.

In Fig. 1, the zonal mass stream-function is generated by vertically integrated zonal wind anomalies from 0 to 1000 hPa pressure level averaged between 5°S and 5°N derived from the lag-regression analysis. The zonal mass stream-function is mainly used to illustrate the direction of anomalous Walker Cells.

## Data availability

All observed and reanalysis data used in this study are publicly available. The HadISST datasets used in this study are available from https://www.metoffice.gov.uk/hadobs/hadisst/data/download.html. The ERA5 atmospheric reanalysis used in this study is available from https://www.ecmwf.int/en/forecasts/datasets/reanalysis-datasets/era5. The ERA-interim atmospheric reanalysis used in this study is available from https://www.ecmwf.int/en/forecasts/datasets/reanalysis-datasets/era-interim. The JRA-55 atmospheric reanalysis used in this study is available from https://jra.kishou.go.jp/JRA-55/index_en.html. The CESM simulation datasets generated and analyzed

in this study are available from S.L. upon request. Source data for generating all the figures in the study have been deposited in the Zenodo database under the accession code https://doi.org/10.5281/zenodo.7902855.

## Code availability

The CESM1.1 source code is available at the website: https://www2.cesm.ucar.edu/models/cesm1.1/. Codes to reproduce the figures are available from S.L. upon request.

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

## Acknowledgements

P.C. and S.G.Y. were partially supported by the Department of Commerce Grant NA20OAR4310408. X.W. was supported by the National Natural Foundation of China (NSFC) project (42276007). The Chinese Scholarship Council (No. 201706330118) is acknowledged for the financial support to S.L. during her visit to Texas A&M University in 2018–2020 when this research was conducted. We thank Z.D. Liu (at the Ocean University of China) and F. Castruccio (at the National Center for Atmospheric Research) for assistance with experimental design.

## Author contributions

P.C. conceived the study and interpreted the results. S.L. conducted CESM simulations, data analysis and visualization. S.L. wrote the initial manuscript with P.C. X.W. contributed to funding acquisition, reviewing, and revising the manuscript. I.R. and S.G.Y. reviewed and improved the manuscript.

## Competing interests

The authors declare no competing interests.
