## [Peer Review File · Nature Communications]

Role of the Maritime Continent in the remote influence of Atlantic Niño on the PacificREVIEWER COMMENTS

Reviewer #1 (Remarks to the Author):

The authors use a coupled model to explore the sensitivity of the Atlantic Niño teleconnection forcing a La Niña response in the tropical Pacific to both orography and land friction in the Maritime Continent (MC).

Through both observations and ensemble simulations, the authors show the importance of atmospheric Kelvin waves in setting up the teleconnection from the Atlantic to the Pacific and then, performing sensitivity studies, they test the role of MC orography and land friction in controlling the strength of the teleconnection.

The study is well designed, thorough, interesting and the manuscript is clearly written, guiding the reader towards its main conclusions. The methods section and the Supplementary figures cover all necessary details on datasets, analysis and experiments.

I guess my only main comment/question is on the sensitivity simulations, NTOP and NFRC. They are identical to the Control simulation in terms of initial conditions and using the same pacemaker methodology, but the orography is removed over the MC in the first one and friction is also artificially reduced in the second one. Given the short integration length, I wonder what are the climatological implications in both simulations. Would the regional SST, moisture transport, precipitation and Walker Circulation still be adjusting to those drastic land surface changes? Perhaps the climate adjustment is removed when comparing [(NTOP+)-(NTOP-)] - [(CTRL+)-(CTRL-)], for example, but that requires strong linearity in the response.

Specific comments:

- line 130: I am not sure I can 'see' a westward propagating Rossby wave in Fig.S5, and I wonder how the authors have drawn the green dashed line. It is somehow more intuitive for the Kelvin waves looking at the geopotential height anomalies contours.

- line 180: Same as above for Fig.3. In May for NTOP, the large geopotential height anomaly seems to be centered at 120E, just as in CTRL. However, for NFRC it is clear that the anomalies propagate further into the Pacific, so it might be a question of redrawing the figure so that anomalies are clearly highlighted? Why not using NTOP-CTRL as in the previous figures?

- I would add in the caption, when appropriate, that the colour scale is different for the CTRL and NTOP-CTRL, NFRC-NTOP panels.

Reviewer #2 (Remarks to the Author):

This is an interesting paper assessing the remote Atlantic effect on ENSO through changes in the Walker circulation, specifically how the ascending branch of the Pacific Walker circulation is modulated. Through a pacemaker-type large ensemble, the authors suggest that equatorial Atlantic SSTs induce eastward propagating Kelvin waves, which interact with Maritime land-atmosphere effects triggering changes in the position of the Pacific branch of the Walker Circulation. As a result, a different type of ENSO may emerge over the tropical Pacific based on the land-atmospheric effects of the maritime continent. They suggest better depiction of land in climate models to better represent the Atlantic influence on ENSO. I thought the authors succinctly explained their results, and their figures represent their findings well. Overall, the paper is well written, and I simply have some minor suggestions and thoughts to be considered. One question that may need to be clarified: Based on the model description stating a pacemaker design, I wonder, does the prescribed constant Atlantic SST anomaly interact with the subsurface ocean in the fully-coupled configuration, and does this affect the atmospheric response at all?

Comments

1. L54: I thought ENSO only occurs in the tropical Pacific. Consider editing word choice in the first sentence of the second paragraph. Perhaps something like, "Although ENSO occurs in the tropical Pacific, modes of climate variability outside of the tropical Pacific contribute to and interact with ENSO, influencing its onset, evolution, and termination."
2. L57: El Niño is the warm phase of ENSO, and I thought the Atlantic Niño is the mode of variability in the Atlantic? Consider changing wording of sentence L56-59.
3. Fig S2 and Fig 1: Averaging two different periods?
4. L82-108: "Regressing out" or partial correlation assumes that ENSO and Atlantic-Niño are linearly dependent, which I doubt is the case. Please clarify.
5. L129 and Fig. S5: Is it possible the slow westward propagating Rossby wave has an effect on the development of an EP-ENSO event, such as in Fig S7f? How do the authors consider the second component of the Matsuno-Gill solution to Atlantic SSTs?
6. When the authors state "local walker Cell over the WP" are they simply referring to the Pacific branch of the Global Walker Circulation, in the traditional sense from Gilbert Walker?

Review by Zachary F. Johnson

Reviewer #3 (Remarks to the Author):

Review of "Role of the Maritime Continent in the remote influence of Atlantic Niño on the Pacific" NCOMMS-22-31374-T by Siying Liu, Ping Chang, Xiuquan Wan, Stephen G. Yeager, Ingo Richter

General comments:

The manuscript investigates the role of the Maritime Continent (MC) on the Atlantic-Pacific teleconnection. The authors argue that the orography of the MC influences the eastward propagating atmospheric Kelvin wave that links the Atlantic variability to the Pacific. The manuscript suggests that improving the representation of land-atmosphere-ocean interactions over the MC may be fundamental to realistically simulate the impact of Atlantic Niño (Niña) on Pacific La Niña (El Niño). The work is well written and presents interesting results. Therefore, I recommend minor revisions.

Although the authors point out that "...the study is not on the debate of how significant an impact Atlantic Niño has on ENSO and its predictability", I think it is important to note whether the CTRL experiment best reproduces the observed Atlantic-Pacific teleconnection. Thus, you can argue that models with a poor representation of the MC orography would overestimate or underestimate the response of the Pacific to the Atlantic forcing.

Why do you consider persisted JJA ATL3 index for the three seasons (FMA, MJJ and ASO) in the experiments? Why not use the spatial patterns of the regression of ATL3 in each of the three seasons FMA, MJJ and ASO? Have you also analyzed the impact of SST anomalies in NDJ prior to ENSO?

Specific comments:

Line 38: "affecting" means reducing or increasing the development of the La Niña?

Line 41: You could use the term "fundamental" instead of "critical".

Line 60-61: Anything about La Niña in 2020 and its strength?

Line 89: "Atlantic Niño begins to develop..." References?

Why do you use MJJ ATL3 instead of JJA ATL3, which is the mature phase of Atlantic Niño?

Line 90-91: "...the remotely forced response...". Although we have some correlation between the FMA and MJJ ATL3 SST anomalies, is it correct to say that we have a response of Pacific to Atlantic in FMA (since Pacific is leading Atlantic in Figure 1a)? I think it would be interesting to also show a figure with the FMA ATL3 before drawing this conclusion.

Figure 1: Why not also show the results of the experiments for the following November-January season, which corresponds to the mature phase of ENSO?

Line 92-93: Have you also tested the results when the Pacific autocorrelation is removed (i.e., the FMA ENSO persistence)?

Legend of Figure 1: Suggested wording for the sentence: "The colors (Figure a-c) and vectors (Figure a-f) indicate statistical significance at the 10% level based on a two-sided Student's t-test."

Line 154: "The accumulated easterlies....". It's really easterlies?

Reply to Reviewer #1

First of all, we would like to thank the reviewer for the constructive comments. The following is our point-to-point reply to each of the comments/questions.

I guess my only main comment/question is on the sensitivity simulations, NTOP and NFRC. They are identical to the Control simulation in terms of initial conditions and using the same pacemaker methodology, but the orography is removed over the MC in the first one and friction is also artificially reduced in the second one. Given the short integration length, I wonder what are the climatological implications in both simulations. Would the regional SST, moisture transport, precipitation and Walker Circulation still be adjusting to those drastic land surface changes? Perhaps the climate adjustment is removed when comparing [(NTOP+)-(NTOP-)] - [(CTRL+)-(CTRL-)], for example, but that requires strong linearity in the response.

Response: We thank the reviewer for their considered comments. The question of whether nonlinearity can influence our analyses and results is excellent but cannot be fully addressed without conducting additional experiments. Therefore, we took the time and effort to conduct a new ensemble of 60 runs, which is identical to the NTOP ensemble except that the prescribed Atlantic Niño SST forcing was removed. We refer to this new ensemble as NTOPclim. The table below summarizes all the ensemble experiments.

Experiment Name	Tropical Atlantic	Maritime Continent Land grid-point	Number of members	Integration Length	Initial Condition
CTRL+/-		Realistic topography	60/60		
NTOP+/-	Model climatology SST plus +/- Atlantic Niño	Flatten topography	60/60	1yr (from Jan)	Branch from the year 2006 of the PI-CTRL simulation
NFRC+/-	SST anomalies	Flatten topography and reduced land friction	60/60		
NTOPclim	No Atlantic Niño SST	Flatten topography	60	1 yr (from Jan)	

With the new ensemble NTOPclim, we can now attempt to evaluate the degree of nonlinearity in [(NTOP+)-(NTOP-)] - [(CTRL+)-(CTRL-)] by comparing its results shown in Fig. 2d-f of the manuscript (also shown in Fig. R1a-c below for easy comparisons) to those obtained by $2 * [((NTOP+) - (NTOPclim)) - ((CTRL+) - (CTRL-)) / 2]$ as shown in Fig. R1d-f. If the system were strictly linear, the results from these two analyses would be identical and both would show an eastward shifted La Niña-like response in the Pacific, because the climatological response to the topography removal would be completely canceled out. As can be seen in Fig. R1, the response patterns between the two analyses are quantitatively similar in the Pacific: both show a reduced cooling (warm anomaly) in the western Pacific and an enhanced cooling (cold anomaly) in the eastern equatorial Pacific, particularly during ASO. Quantitatively, however, there are some differences between these results in the sense that $2 * [((NTOP+) - (NTOPclim)) - ((CTRL+) - (CTRL-)) / 2]$ produces a stronger and more consistent eastward shift of La Niña-like response after the MC topography is flattened. In addition, the responses over the Indian Ocean in NTOP-CTRL (Fig. R1a-c) are stronger than that in Fig. R1d-f, which is not influencing the main point in this study. This result suggests that [(NTOP+)-(NTOP-)] - [(CTRL+)-(CTRL-)] may underestimate the eastward shift of the La Niña-like response because of the relatively weak nonlinear interaction between the climatological state changes due to the MC topography removal and Atlantic Niño SST forced response. We further note that the results from $2 * [((NTOP+) - (NTOPclim)) - ((CTRL+) - (CTRL-)) / 2]$

Fig R1. **a-c** Simulated SST ($^{\circ}\text{C}$; color shading) and winds at 992 hPa (m s^{-1} ; vectors) mean during Feb-Mar-Apr (FMA) (**a**), May-Jun-Jul (MJJ) (**b**) and Aug-Sep-Oct (ASO) (**c**) respectively by NTOP minus CTRL. **d-f** Same with **a-c** except for $2*[(\text{NTOP}+ \text{ minus } \text{NTOPclim}) - (\text{CTRL}+ \text{ minus } \text{CTRL-})/2]$. The colors and vectors indicate statistical significance at the 10% level based on a two-sided Student's t-test.

$[(\text{CTRL}+) - (\text{CTRL-})]/2]$ (Fig. R1d-f) are more consistent with those in NFRC-NTOP shown in Fig.2g-i of the manuscript. Therefore, we conclude that the relatively weak nonlinearity in $[(\text{NTOP}+) - (\text{NTOP-})] - [(\text{CTRL}+) - (\text{CTRL-})]$ should not fundamentally alter the conclusion of the study. We expect a similar conclusion holds for NFRC-NTOP, as the response pattern is more robust. We added some discussions in the text and the Supplementary Discussion to discuss these results. Specifically, the following sentences are added to Section 2.2 (L218-223):

“We note that the analysis used here to compare these different ensembles implicitly assumes weak nonlinearity in the system, such that the climatological state changes due to the modification of the MC orography and land friction in NTOP and NFRC do not introduce strong nonlinear effects on the response to Atlantic Niño SST. We present additional analyses in the Supplementary Discussion that validate the weak nonlinearity assumption.”

A discussion is also added to Supplementary Information (Supplementary Discussion):

“Supplementary Discussion

To validate the weak nonlinearity assumption, we conduct a new ensemble of 60 runs, which is identical to the NTOP ensemble except that the prescribed Atlantic Niño SST forcing was removed. We refer to this new ensemble as NTOPclim. We evaluate the degree of nonlinearity in $[(\text{NTOP}+) - (\text{NTOP-})] - [(\text{CTRL}+) - (\text{CTRL-})]$ by comparing its results shown in Fig. 2d-f (also in Supplementary Fig. 14a-c for easy comparisons) to those obtained by $2*[(\text{NTOP}+) - (\text{NTOPclim})] - [(\text{CTRL}+) - (\text{CTRL-})]/2]$ as shown in Supplementary Fig.14d-f. If the system were strictly linear, the results from these two

analyses would be identical and both would show an eastward shifted La Niña-like response in the Pacific, because the climatological response to the topography removal would be completely canceled out. As can be seen in Supplementary Fig. 14, the response patterns between the two analyses are qualitatively similar, both show a reduced cooling (warm anomaly) in the western Pacific and an enhanced cooling (cold anomaly) in the eastern equatorial Pacific, particularly during ASO. Quantitatively, however, there are some differences between these results in the sense that $2*[(\text{NTOP+}) - (\text{NTOPclim})] - ((\text{CTRL+}) - (\text{CTRL-})) / 2$ produces a stronger and more consistent eastward shift of La Niña-like response after the MC topography is flattened. This result suggests that $[(\text{NTOP+}) - (\text{NTOP-})] - [(\text{CTRL+}) - (\text{CTRL-})]$ may underestimate the eastward shift of La Niña-like response because of the relatively weak nonlinear interaction between the climatological state changes due to the MC topography removal and Atlantic Niño SST forced response. We further note that the results from $2*[(\text{NTOP+}) - (\text{NTOPclim})] - ((\text{CTRL+}) - (\text{CTRL-})) / 2$ (Supplementary Fig. 14d-f) are more consistent with those in NFRC-NTOP shown in Fig.2g-i. Therefore, we conclude that the relatively weak nonlinearity in $[(\text{NTOP+}) - (\text{NTOP-})] - [(\text{CTRL+}) - (\text{CTRL-})]$ should not fundamentally alter the conclusion of the study. We expect a similar conclusion holds for NFRC-NTOP, as the response pattern is more robust.”

line 130: I am not sure I can `see' a westward propagating Rossby wave in Fig.S5, and I wonder how the authors have drawn the green dashed line. It is somehow more intuitive for the Kelvin waves looking at the geopotential height anomalies contours.

Fig R2. a-b Hovmöller diagrams of simulated meridional mean geopotential height at 200 hPa six-hourly anomalies (a) (m; color shading and contours) averaged from 15°S to 5°S and zonal wind daily anomalies at 850 hPa (b) (m s⁻¹; color shading) averaged from 10°S to 0° along the equator by CTRL+ minus CTRL-.

Response: Thank you for pointing this out. We agree that the westward propagation of Rossby waves is difficult to identify in the original figure. Following your suggestion, we plotted

geopotential height anomalies at 200 hPa in (CTRL+)-(CTRL-) (Fig. R2), which now seem to reveal more clearly an eastward propagating (Kelvin wave-like) and a slower westward propagating (Rossby wave-like) signal. These propagating signals are further supported by the similar features in zonal wind anomalies at 850 hPa in (CTRL+)-(CTRL-). Considering the Atlantic Niño heating source primarily locates near and south of the equator and the structure of the Matsuno-Gill analytical solution, we choose the latitude band of 5°-15°S for the geopotential height anomalies and the band of 0°-10°S for the zonal wind anomalies during the first 60 days in CTRL. We believe these propagating signals are consistent with the Kelvin and Rossby wave solutions in Matsuno-Gill model, and we now replaced the original Supplementary Fig. 5 by this new figure as Supplementary Fig. 8 in the Supplementary Information file.

line 180: Same as above for Fig.3. In May for NTOP, the large geopotential height anomaly seems to be centered at 120E, just as in CTRL. However, for NFRC it is clear that the anomalies propagate further into the Pacific, so it might be a question of redrawing the figure so that anomalies are clearly highlighted? Why not using NTOP-CTRL as in the previous figures? I would add in the caption, when appropriate, that the colour scale is different for the CTRL and NTOP-CTRL, NFRC-NTOP panels.

Response: We appreciate this excellent suggestion. Following your suggestion, we plotted the geopotential height anomalies using NTOP-CTRL and NFRC-NTOP (Fig. R3). As expected, these difference plots do show a further eastward expansion of the anomalies in NTOP and NFRC, but they also add some noise to the response patterns, particularly in the off-equatorial regions. Therefore, we decide to present these results in the Supplementary Fig. 11 in the Supplementary Information file.

Fig R3. **a-c** Simulated geopotential height anomalies at 200 hPa (m; color shading) in March (Mar) (**a**), April (Apr) (**b**), May (**c**) for NTOP minus CTRL. Pots represent those that are significant at the 10% level based on a two-sided Student's t-test. **d-f** Same as **a-c** except for NFRC minus CTRL. Anomalies in CTRL, NTOP and NFRC are obtained by taking ensemble-mean of CTRL+ minus CTRL-, ensemble-mean of NTOP+ minus NTOP-, and ensemble-mean of NFRC+ minus NFRC-, respectively.

Reply to Reviewer #2

First of all, we would like to thank the reviewer for the helpful and constructive comments. The following is our point-to-point reply to each of the comments.

One question that may need to be clarified: Based on the model description stating a pacemaker design, I wonder, does the prescribed constant Atlantic SST anomaly interact with the subsurface ocean in the fully-coupled configuration, and does this affect the atmospheric response at all?

Response: We thank the reviewer to raise this question. In all our simulations where Atlantic Niño SST anomalies were used as a forcing, we used a restoring technique in which model simulated SST is relaxed towards the constant Atlantic Niño-like SST anomalies over the region between 69°W-18°E and 24°S-24°N with a restoring timescale of 5 days. With such a restoring, the simulated SST within the forcing region is heavily constrained by the specified SSTs, although it can still vary to some degree. The subsurface ocean, however, is not constrained, and thus can respond to the wind changes driven by the Atlantic Niño SST anomalies. Therefore, one expects to see a positive (negative) ocean heat content anomaly develop in the eastern equatorial Atlantic when the positive (negative) Atlantic Niño SST is prescribed. This change in the ocean heat content can feedback onto the simulated equatorial Atlantic SSTs, which in turn can affect the Atlantic Walker Cell via local Bjerknes feedback. However, the magnitude of this feedback is expected to be relatively small because of the strong SST restoring. This is confirmed by Fig. R4 (which has now been added to the Supplementary Information as Supplementary Fig. 7) that shows a Hovmöller diagram of SST along the equator.

Fig R4. Hovmöller diagram of SST (°C; color shading) and ocean heat content (10^{20} Joule; contours) anomalies in upper 200 meters in CTRL (CTRL+ minus CTRL-) from January to December averaged from 2°S to 2°N along the equator.

As can be seen, there is some strengthening of the equatorial Atlantic SST during the boreal summer induced by the large ocean heat content increase in the eastern equatorial Atlantic. This

phase-locking to the summer months is in line with the observations (Richter et al. 2017, ¹⁹ in the manuscript). Therefore, we expect some contributions of the local Bjerknes feedback in the equatorial Atlantic to the remote influence in the Pacific, particularly during the late boreal spring and summer. We added a brief discussion in the revision to point out that some of the enhanced response over the western Pacific during boreal summer may be attributed to the increase in the Atlantic Niño SST forcing caused by the local Bjerknes feedback. Specifically, we added the following paragraph to Section 2.1 (L132-140):

“It is worth noting that although the simulated SST in the tropical Atlantic is heavily constrained by the prescribed time-invariant Atlantic Niño SST anomalies, there are still weak variations in equatorial Atlantic SST with a peak during boreal summer in agreement with the observations¹⁹. This indicates that the unconstrained subsurface ocean variability in response to Atlantic Niño SST induced trade wind changes can modulate the simulated equatorial Atlantic SST (Supplementary Fig. 7) through a local Bjerknes feedback within the tropical Atlantic. This local Bjerknes feedback in the Atlantic may contribute to the enhanced remote response in the Pacific during boreal summer.”

However, we do not expect the equatorial Atlantic subsurface ocean variability to have a strong influence on other ocean basins through oceanic teleconnections in the simulations, because of the short simulation length (only one year). Oceanic teleconnections take longer than one year to establish.

L54: I thought ENSO only occurs in the tropical Pacific. Consider editing word choice in the first sentence of the second paragraph. Perhaps something like, “Although ENSO occurs in the tropical Pacific, modes of climate variability outside of the tropical Pacific contribute to and interact with ENSO, influencing its onset, evolution, and termination.”

Response: Thanks for the comment, and we made the change accordingly in L55-57.

L57: El Niño is the warm phase of ENSO, and I thought the Atlantic Niño is the mode of variability in the Atlantic? Consider changing wording of sentence L56-59.

Response: Thanks for the comments, and we made the change accordingly (L57-60). It reads:

“In particular, studies have shown that Atlantic Niño or Atlantic Zonal Mode⁵ – a phenomenon in the equatorial Atlantic similar to ENSO, albeit weaker amplitude and more sporadic, can remotely force a La Niña-like response in the Pacific⁶⁻¹⁶.”

Fig S2 and Fig 1: Averaging two different periods?

Response: Thank you for the question. Yes, to fully utilize the three reanalysis products and minimize internal variability, we first averaged the wind fields from the three reanalysis datasets and then performed the lag-regression analysis (Fig. 1 and Supplementary Fig. 2, 10). Since the time periods of the reanalysis products are different, we used all three datasets in the common time period of 1979-2018 and only used ERA5 and JRA-55 datasets for the remaining periods to derive

the averaged wind fields. We have now made this clear in the **Methods** of the manuscript (L376-382):

“To fully utilize the three reanalysis products and minimize internal variability, we first averaged the wind fields from the three reanalysis datasets and then performed the lag-regression analysis (Fig. 1 and Supplementary Fig. 2, 10). Since the time periods of the reanalysis products are different, we used all three datasets in the common time period of 1979-2018 and only used ERA5 and JRA-55 datasets for the remaining periods to derive the averaged wind fields.”

L82-108: “Regressing out” or partial correlation assumes that ENSO and Atlantic-Niño are linearly dependent, which I doubt is the case. Please clarify.

Response: Yes, we agree that ENSO and Atlantic-Niño are not necessarily linearly dependent, which casts some doubts on the partial correlation analysis. For this reason, we also carried out another analysis by excluding strong ENSO years (rather than by linearly regressing out ENSO). This method minimizes the potential nonlinear effect of preexisting strong ENSO events on the Atlantic Niño remote influence, but at the expense of shortening the available data record, leading to a noisier response pattern as shown in Supplementary Fig. 2. However, the salient feature of the response remains unchanged as shown in Supplementary Fig. 2. Therefore, we believe that the result of partial correlation used by many previous studies in identifying the remotely forced response is robust.

Of course, one can still argue that Atlantic-Niño itself may be a response to ENSO in the first place, but this is not the question that we are addressing in this study. The question we attempt to address is: Given an Atlantic Niño SST forcing (regardless of its origin), can it have a remote influence on Pacific? We believe our numerical experiments provide further supports to our observational analysis results.

L129 and Fig. S5: Is it possible the slow westward propagating Rossby wave has an effect on the development of an EP-ENSO event, such as in Fig S7f? How do the authors consider the second component of the Matsuno-Gill solution to Atlantic SSTs?

Response: You have raised an interesting and important question here. We did examine the possibility of the slow westward propagating Rossby wave forcing an EP-ENSO event. We note that the Rossby wave component of the Matsuno-Gill solution generates a westerly wind anomaly (in contrast to the easterly anomaly generated by the Kelvin wave). The westerly wind anomaly can reach to the EP, possibly producing a brief warming in the far eastern equatorial Pacific (see Fig. 2a). But such a forcing is too weak to compete with the Kelvin wave induced easterly anomalies and the subsequent Bjerknes feedback. The initial warming signal induced by the westerly anomaly of the Rossby wave appears to be quickly overcome by the strong equatorial cooling associated with the easterly wind anomaly developed in the wake of Kelvin wave. One possible reason for the weak Rossby wave forcing is that the wind anomalies associated with the Rossby wave have maximum amplitude off the equator, and thus is not effective in triggering Bjerknes feedback. We have made some interpretation in the Manuscript in L162-172. It reads:

“The Atlantic Niño SST forcing also generates a Rossby wave component of the Matsuno-Gill solution that carries a weak westerly wind anomaly westward (in

contrast to the easterly anomaly of the Kelvin wave) (Supplementary Fig. 8). During boreal spring, a brief warming in the far eastern equatorial Pacific may be associated with the Rossby wave (Fig. 2a). But apparently this forcing is too weak to compete with the easterly anomalies induced by the Kelvin wave and the subsequent Bjerknes feedback. The initial warming appears to be quickly overcome by the strong equatorial cooling associated with the easterly wind anomaly developed in the wake of Kelvin wave. One possible reason for the weak Rossby wave forcing is that the wind anomalies associated with the Rossby wave have maximum amplitude off the equator (Fig. 1b and Fig. 2b), and thus is not effective in triggering Bjerknes feedback.”

When the authors state “local walker Cell over the WP” are they simply referring to the Pacific branch of the Global Walker Circulation, in the traditional sense from Gilbert Walker?

Response: We appreciate you bring out this point. In the manuscript, the “local walker cell over the WP” refers to the anomalous zonal-vertical circulation over the WP in response to the Atlantic Niño. This anomalous cell acts to strengthen the climatological Walker Circulation (in the traditional sense from Gilbert Walker), which leads to development of the La Niña-like response in the Pacific.

Reply to Reviewer #3

We thank the reviewer for the thorough evaluation of our paper, and the helpful suggestions and comments, which have significantly improved the manuscript. Below are our point-to-point responses to each comment.

Although the authors point out that “...the study is not on the debate of how significant an impact Atlantic Niño has on ENSO and its predictability”, I think it is important to note whether the CTRL experiment best reproduces the observed Atlantic-Pacific teleconnection. Thus, you can argue that models with a poor representation of the MC orography would overestimate or underestimate the response of the Pacific to the Atlantic forcing.

Response: Thank you for the comment, which is well taken. In the revision, we have highlighted the importance of realistically simulating the observed Atlantic-Pacific teleconnection in CTRL by adding the sentence to Section 2.1, “It is important that CTRL can realistically reproduce the observed Atlantic-to-Pacific teleconnection, because it provides not only support to the observational analysis, but also a reference for the mechanistic model simulations described below.” (L129-131). We also added another sentence to Section 3 to highlight the potential impact of poorly resolved orography over MC in climate models. It reads “However, the results presented here show that a poorly resolved MC orography may have a direct impact on simulations of the observed Pacific response to the Atlantic forcing that may be important to improving ENSO-related climate prediction.” (L256-259).

Why do you consider persisted JJA ATL3 index for the three seasons (FM, MJJ and ASO) in the experiments? Why not use the spatial patterns of the regression of ATL3 in each of the three seasons FMA, MJJ and ASO?

Response: We did conduct similar ensembles of simulations where tropical Atlantic SST was restored to a time-varying Atlantic Niño SST pattern. The spatial pattern of the SST forcing is

Fig R5. a-c Simulated SST (°C; color shading) and wind anomalies at 992hPa (m s⁻¹; arrows) averaged over boreal spring (FMA) (a), summer (MJJ) (b) and fall (ASO) (c) in CTRLT. Anomalies in CTRLT are obtained by taking ensemble mean of CTRLT+ minus CTRLT-. The colors and vectors indicate statistical significance at the 10% level based on a two-sided Student's t-test.

identical to that in CTRL, but we ramped up its amplitude gradually from zero to a maximum during boreal summer, and then to zero at the end of the simulation according to a half sine function. The ensemble size of these simulations is 40 for each positive and negative Atlantic Niño-like SST forcing. These ensembles are referred to as CTRLT+ and CTRLT-. Fig. R5 shows the results of these simulations constructed in the same way as those of CTRL+ - CTRL-. As can be seen, the results are very similar to those in CTRL+ - CTRL-, and both reproduce very well the observed results shown in Fig. 1a-c. The only difference between CTRL and CTRLT lies in the strength of the response over the Pacific with CTRL producing a stronger response as expected. Given the similarity of these results and emphasis of this study on the underlying mechanism of the tropical Atlantic-Pacific teleconnection, we decided to use CTRL to cleanly dissect the response signal to the Atlantic Niño.

We would like to point out that even though we restored model SSTs to a time-invariant Atlantic Niño-like SST anomaly in the tropical Atlantic, there are still time variations in simulated tropical Atlantic SSTs in CTRL. This is because the subsurface ocean is not constrained and can vary in response to wind changes induced by the specified Atlantic Niño SST forcing. These subsurface ocean changes can modulate the SST through local Bjerknes feedback (see our reply to comment#1 by reviewer#2). Therefore, even though we used a time-invariant SST anomaly, the simulated tropical Atlantic SST can vary in time with a peak in boreal summer as in observations (see Fig. R4 or Supplementary Fig. 7 in our reply to reviewer#2).

Have you also analyzed the impact of SST anomalies in NDJ prior to ENSO?

Response: No, we did not analyze the impact of NDJ Atlantic SST anomalies on the Pacific if we understood your question correctly. This is because Atlantic Niño related SST anomalies are relatively weak during NDJ as shown by many previous studies. Okumura and Xie (2006)¹ identified a second peak of Atlantic Niño during ND and referred to it as Atlantic Niño II, but its standard deviation is about half of that of the Atlantic Niño I during MJJ and also it lasts for a shorter period of time. Since Atlantic Niño SST forcing is already argued to be too weak to have a significant impact on the Pacific, we decided not to focus on the weaker Atlantic Niño II.

However, we did try to regress out the NDJ Niño3.4 index in our partial regression analysis, in case that your question is in regards to the impact of regressed out ENSO signal. Fig. R6 shows that regressing out NDJ Niño3.4 in the partial regression analysis results in a nearly identical response to Atlantic Niño to that of regressing out DJF Niño3.4 in our original analysis shown in Fig. 1.

Fig R6. **a-c** Boreal spring (February-March-April (FMA)) **(a)**, summer (May-June-July (MJJ)) **(b)**, and fall (August-September-October (ASO)) **(c)** SST ($^{\circ}\text{C}$; color shading) anomalies (HadISST, 1959-2021) and wind anomalies at 1000hPa (m s^{-1} ; color arrows) averaged using three reanalysis datasets (ERA5, JRA55 for 1959-2021 and ERA-Interim for 1979-2018) and regressed onto the normalized May-June-July (MJJ) ATL3 index after linearly regressing out NDJ Niño3.4. **d-f** Same as **a-c** except for zonal and vertical wind anomalies (m s^{-1} ; arrows; vertical winds multiplied by 100) averaged between 5°S and 5°N , and zonal mass stream-function anomalies (Kg s^{-1} ; color shading; see Methods) from 175 to 1000 hPa pressure levels. The colors (Fig a-c) and vectors (Fig a-f) indicate statistical significance at the 10% level based on a two-sided Student's t-test. The statistical test was not performed for the zonal mass stream-function because its main purpose is to illustrate the direction of the zonal circulation anomalies.

Line 38: “affecting” means reducing or increasing the development of the La Niña?

Response: Thank you for the question. Exactly, the phrasing here can be understood as reducing or increasing the development of the La Niña. More specifically, during the development of an ENSO event, the strength of the Bjerknes feedback can affect the speed and amplitude of the growth in SSTA and east-west SST gradient changes.

Line 41: You could use the term “fundamental” instead of “critical”.

Response: Thanks. The change has been done in Line 41.

Line 60-61: Anything about La Niña in 2020 and its strength?

Response: Thank you for the suggestion. Fig. R7 shows the evolution of Pacific SST and wind anomalies from FMA 2020 to NDJ 2021 using the HadISST and ERA5 Reanalysis datasets. Interestingly, there was a warm anomaly over the tropical Atlantic during FMA 2020 (but not exactly in the form of Atlantic Niño) and then a weaker warm Atlantic anomaly during MJJ 2020 before the development of the strong La Niña event during NDJ 2020/21. The strong Atlantic Niño of 2021 appeared in MJJ 6 months before the strong La Niña in NDJ 2021/22. Since the warming pattern in the tropical Atlantic during boreal spring and summer 2020 does not have an appearance of Atlantic Niño, it is not clear if there is a connection between the warming pattern in tropical Atlantic during boreal spring and summer 2020 and 2020/21 La Niña in the Pacific. It is also possible that the 2021 Atlantic Niño is a response to the 2020/21 La Niña in the Pacific. All these questions deserve future studies.

Fig R7. a-h Seasonal mean of the observed SST and wind vectors at 1000 hPa anomalies for 2020 FMA (Feb-Mar-Apr) (a), 2020 MJJ (May-Jun-Jul) (b), 2020 ASO (Aug-Sep-Oct) (c), 2020 NDJ¹ (Nov-Dec-Jan) (d), 2021 FMA (Feb-Mar-Apr) (e), 2021 MJJ (May-Jun-Jul) (f), 2021 ASO (Aug-Sep-Oct) (g), and 2021 NDJ¹ (h) respectively using HadISST and ERA5 Reanalysis. The climatology mean is calculated by the mean from 1959 to 2021.

Line 89: “Atlantic Niño begins to develop...” References?

Response: Citation has now been added in Line 90 and we would like to thank you for the comment. As suggested by Richter et al. (2017) (¹⁹ in the manuscript) using composite analyses of

observations, Atlantic Niño related SSTA typically starts to develop in the central equatorial Atlantic during March (see Fig. R8 taken from Fig. 3 of Richter et al. (2017)).

Fig R8. (Taken from Fig. 3 in Richter et al. (2017)) OI SST anomalies (shading; K), ERAI surface wind anomalies (vectors; reference 1 m/s), and total precipitation (contour lines; interval 3 mm/day) composited on positive AZM events (canonical Atlantic Niño).

Why do you use MJJ ATL3 instead of JJA ATL3, which is the mature phase of Atlantic Niño?

Response: Based on the composite analysis of observed Atlantic Niño events, Atlantic Niño related SSTA reaches the maximum in the ATL3 region during June (see Fig. R8). Therefore, we choose MJJ to better capture the peak of the Atlantic Niño. We added some discussions about MJJ ATL3 pattern used for regression in the **Methods** (L354-357). It reads:

“Note that we did not use the commonly used Jun-Jul-Aug (JJA) ATL3 index in Fig.1, because Atlantic Niño related SST anomalies reach the maximum in the ATL3 region during June based on the composite analysis of observed Atlantic Niño events¹⁹.”

Line 90-91: “...the remotely forced response...”. Although we have some correlation between the FMA and MJJ ATL3 SST anomalies, is it correct to say that we have a response of Pacific to Atlantic in FMA (since Pacific is leading Atlantic in Figure 1a)? I think it would be interesting to also show a figure with the FMA ATL3 before drawing this conclusion.

Response: Thank you for suggesting this additional analysis. The partial linear lag-regression against FMA ATL3 index, as you suggested, is shown in Fig. R9, after regressing out DJF Niño3.4 index. The methodology and observational datasets used are exactly the same as those shown in Fig. 1 of the manuscript, except for the FMA ATL3 index.

Fig R9. **a-c** Boreal spring (February-March-April (FMA)) **(a)**, summer (May-June-July (MJJ)) **(b)**, and fall (August-September-October (ASO)) **(c)** SST ($^{\circ}\text{C}$; color shading) anomalies (HadISST, 1959-2021) and wind anomalies at 1000hPa (m s^{-1} ; color arrows) averaged using three reanalysis datasets (ERA5, JRA55 for 1959-2021 and ERA-Interim for 1979-2018) and regressed onto the normalized Feb-Mar-Apr (FMA) ATL3 index after linearly regressing out DJF Niño3.4. **d-f** Same as **a-c** except for zonal and vertical wind anomalies (m s^{-1} ; arrows; vertical winds multiplied by 100) averaged between 5°S and 5°N , and zonal mass stream-function anomalies (Kg s^{-1} ; color shading; see Methods) from 175 to 1000 hPa pressure levels. The colors (Fig a-c) and vectors (Fig a-f) indicate statistical significance at the 10% level based on a two-sided Student's t-test. The statistical test was not performed for the zonal mass stream-function because its main purpose is to illustrate the direction of the zonal circulation anomalies.

As can be seen from Fig. R9, during FMA there is a significant positive regression over much of the equatorial and south tropical Atlantic along with a local westerly anomaly in the western tropical Atlantic. Over the western Pacific, we again see a strong easterly anomaly coupled with some weak SST anomalies. These regression patterns are not so different from those regressed onto MJJ ATL3 index as shown in Fig. 1a of the manuscript. The patterns in the following seasons are also similar, except that the regressed SSTAs in tropical Atlantic decay in amplitude, as expected. These results provide additional supports to the notion that the patterns in the Pacific are likely a response to Atlantic Niño. This notion is further supported by the numerical experiments presented in this study where no preexisting Pacific anomalies are present and the response in the Pacific can only be attributed to Atlantic Niño. Now we added Fig. R10 to Supplementary Information as Supplementary Fig. 12 and some discussions in the **Methods** (L357-365). It reads:

“The partial linear lag-regression against Feb-Mar-Apr (FMA) ATL3 index is also shown in Supplementary Fig. 12, after regressing out DJF Niño3.4 index. It shows, positive regression over much of the equatorial and south tropical Atlantic along with a westerly anomaly in the western tropical Atlantic during FMA. Over the western Pacific, a strong easterly anomaly coupled with some weak SST anomalies is also detected during FMA. These regression patterns exhibit strong similarity with the regression against MJJ ATL3 index in Fig. 1, except that the regressed SST anomalies in the tropical Atlantic decay faster in amplitude as expected.”

Figure 1: Why not also show the results of the experiments for the following November-January

season, which corresponds to the mature phase of ENSO?

Fig R10. **a-d** Boreal spring FMA (**a**), summer MJJ (**b**), fall ASO (**c**) and winter NDJ (**d**) SST ($^{\circ}\text{C}$; color shading) anomalies (HadISST, 1959-2021) and wind anomalies at 1000hPa (m s^{-1} ; color arrows) averaged using three reanalysis datasets (ERA5, JRA55 for 1959-2021 and ERA-Interim for 1979-2018) and regressed onto the normalized MJJ ATL3 index after linearly regressing out DJF Niño3.4. **e-h** Same as **a-d** except for zonal and vertical wind anomalies (m s^{-1} ; arrows; vertical winds multiplied by 100) averaged between 5°S and 5°N , and zonal mass stream-function anomalies (Kg s^{-1} ; color shading; see Methods) from 175 to 1000 hPa pressure levels. The colors (Fig a-d) and vectors (Fig a-h) indicate statistical significance at the 10% level based on a two-sided Student's t-test. The statistical test was not performed for the zonal mass stream-function because its main purpose is to illustrate the direction of the zonal circulation anomalies.

Response: Thank you for this suggestion. We added the lag-regression analysis in November-January to Fig. R10 as an extension of Fig. 1 in our manuscript using the same observation datasets. We also added the simulated results in CTRL, [NTOP-CTRL], and [NFRC-NTOP] during November-December to Fig. R11. Because the integration length of our simulations is only one year, we used November-December mean in Fig. R11d, h, l. As expected, these results show that the remotely forced La Niña continues to grow because of the basin-wide Bjerknes feedback. These results also show that the model is capable of realistically simulating the observed La Niña-like response in the Pacific from its developing phase to its mature phase. We now include these results to Supplementary Fig. 4, 10, L105-108 and L156-158. It reads:

“In the following fall and winter, as Atlantic Niño is winding down, the positive Bjerknes feedback continues in the tropical Pacific, eventually giving rise to a full-fledged La Niña (Fig. 1c and Supplementary Fig. 4a) coupled with a fully developed anomalous Walker Circulation over the Pacific (Fig. 1f).”

“This is indicative of a basin-wide Bjerknes feedback, which leads to the full development of La Niña during boreal winter (Supplementary Fig. 10a).”

Fig R11. a-d Simulated SST ($^{\circ}\text{C}$; color shading) and wind anomalies at 992hPa (m s^{-1} ; arrows) averaged over boreal spring (FMA) (a), summer (MJJ) (b) fall (ASO) (c) and winter (ND) in CTRL; e-h Same as a-d except for the anomalies in NTOP minus those in CTRL; i-l Same as a-d except for the anomalies in NFRC minus those in NTOP. Anomalies in CTRL, NTOP and NFRC are obtained by taking ensemble mean of CTRL+ minus CTRL-, ensemble mean of NTOP+ minus NTOP-, and ensemble mean of NFRC+ minus NFRC-, respectively. Colors and vectors represent those that are significant at the 10% level based on a two-sided Student's t-test. Note that in e-h and i-l the tropical Atlantic sector is excluded because the forcing is the same for all three experiments.

Line 92-93: Have you also tested the results when the Pacific autocorrelation is removed (i.e., the FMA ENSO persistence)?

Response: As shown in Supplementary Fig. 2, we performed an additional analysis by removing strong ENSO years first and then performing lag-regression analyses. In this analysis, ENSO persistence is removed, and the results are similar with those in Fig. 1. Therefore, we are confident that the results are not sensitive to the methods of removing ENSO. Again, our numerical

experiments provided further supports to our observational analyses.

Legend of Figure 1: Suggested wording for the sentence: "The colors (Figure a-c) and vectors (Figure a-f) indicate statistical significance at the 10% level based on a two-sided Student's t-test."

Response: The legend has now been corrected as suggested in Fig.1 and thank you for pointing this out.

Line 154: "The accumulated easterlies....". It's really easterlies?

Response: Thank you for the clarification question. Easterlies are now changed to easterly anomalies in Line 180.

Reference

1. Okumura, Y. & Xie, S.-P. Some overlooked features of tropical Atlantic climate leading to a new niño-like phenomenon*. *Journal of Climate* 19, 5859–5874 (2006).

REVIEWERS' COMMENTS

Reviewer #1 (Remarks to the Author):

I am fully satisfied with the revised manuscript and response to my queries. I believe this is a very interesting and thorough study that will be very useful to the climate modelling community, and I recommend its publication.

Reviewer #3 (Remarks to the Author):

Review of "Role of the Maritime Continent in the remote influence of Atlantic Niño on the Pacific" NCOMMS-22-31374-T by Siying Liu, Ping Chang, Xiuquan Wan, Stephen G. Yeager, Ingo Richter

The authors have considered the suggestions and responded appropriately to my comments. Therefore, I recommend that the manuscript be accepted for publication and congratulate the authors for the work done.